# HYPRO: A Hybridly Normalized Probabilistic Model for Long-Horizon Prediction of Event Sequences

**Siqiao Xue, Xiaoming Shi, James Y Zhang**
Ant Group
569 Xixi Road
Hangzhou, China
siqiao.xsq@alibaba-inc.com
{peter.sxm,james.z}@antgroup.com

**Hongyuan Mei**
Toyota Technological Institute at Chicago
6045 S Kenwood Ave
Chicago, IL 60637
hongyuan@ttic.edu

## Abstract

In this paper, we tackle the important yet under-investigated problem of making long-horizon prediction of event sequences. Existing state-of-the-art models do not perform well at this task due to their autoregressive structure. We propose HYPRO, a hybridly normalized probabilistic model that naturally fits this task: its first part is an autoregressive base model that learns to propose predictions; its second part is an energy function that learns to reweight the proposals such that more realistic predictions end up with higher probabilities. We also propose efficient training and inference algorithms for this model. Experiments on multiple real-world datasets demonstrate that our proposed HYPRO model can significantly outperform previous models at making long-horizon predictions of future events. We also conduct a range of ablation studies to investigate the effectiveness of each component of our proposed methods.

## 1   Introduction

Long-horizon prediction of event sequences is essential in various real-world applied domains:

- *Healthcare.* Given a patient's symptoms and treatments so far, we would be interested in predicting their future health conditions *over the next several months*, including their prognosis and treatment.
- *Commercial.* Given an online consumer's previous purchases and reviews, we may be interested in predicting what they would buy *over the next several weeks* and plan our advertisement accordingly.
- *Urban planning.* Having monitored the traffic flow of a town for the past few days, we'd like to predict its future traffic *over the next few hours*, which would be useful for congestion management.
- Similar scenarios arise in *computer systems*, *finance*, *dialogue*, *music*, etc.

Though being important, this task has been under-investigated: the previous work in this research area has been mostly focused on the prediction of the *next single* event (e.g., its time and type).

In this paper, we show that previous state-of-the-art models suffer at making long-horizon predictions, i.e., predicting *the series of future events over a given time interval*. That is because those models are all *autoregressive*: predicting each future event is conditioned on all the previously predicted events; an error can not be corrected after it is made and any error will be cascaded through all the subsequent predictions. Problems of the same kind also exist in natural language processing tasks such as generation and machine translation (Ranzato et al., 2016; Goyal, 2021).

In this paper, we propose a novel modeling framework that learns to make long-horizon prediction of event sequences. Our main technical contributions include:

- *A new model.* The key component of our framework is HYPRO, a hybridly normalized neural probabilistic model that combines an autoregressive base model with an energy function: the

36th Conference on Neural Information Processing Systems (NeurIPS 2022).

base model learns to propose plausible predictions; the energy function learns to reweight the proposals. Although the proposals are generated autoregressively, the energy function reads each entire completed sequence (i.e., true past events together with predicted future events) and learns to assign higher weights to those which appear more realistic *as a whole*.

Hybrid models have already demonstrated effective at natural language processing such as detecting machine-generated text (Bakhtin et al., 2019) and improving coherency in text generation (Deng et al., 2020). We are the first to develop a model of this kind for time-stamped event sequences. Our model can use any autoregressive event model as its base model, and we choose the state-of-the-art continuous-time Transformer architecture (Yang et al., 2022) as its energy function.

- *A family of new training objectives.* Our second contribution is a family of training objectives that can estimate the parameters of our proposed model with low computational cost. Our training methods are based on the principle of noise-contrastive estimation since the log-likelihood of our HYPRO model involves an intractable normalizing constant (due to using energy functions).

- *A new efficient inference method.* Another contribution is a normalized importance sampling algorithm, which can efficiently draw the predictions of future events over a given time interval from a trained HYPRO model.

## 2 Technical Background

### 2.1 Formulation: Generative Modeling of Event Sequences

We are given a fixed time interval $[0, T]$ over which an event sequence is observed. Suppose there are $I$ events in the sequence at times $0 < t_1 < \ldots < t_I \leq T$. We denote the sequence as $x_{[0,T]} = (t_1, k_1), \ldots, (t_I, k_I)$ where each $k_i \in \{1, \ldots, K\}$ is a discrete event type.

Generative models of event sequences are temporal point processes. They are **autoregressive**: events are generated from left to right; the probability of $(t_i, k_i)$ depends on the history of events $x_{[0,t_i)} = (t_1, k_1), \ldots, (t_{i-1}, k_{i-1})$ that were drawn at times $< t_i$. They are **locally normalized**: if we use $p_k(t \mid x_{[0,t)})$ to denote the probability that an event of type $k$ occurs over the infinitesimal interval $[t, t+dt)$, then the probability that nothing occurs will be $1 - \sum_{k=1}^{K} p_k(t \mid x_{[0,t)})$. Specifically, temporal point processes define functions $\lambda_k$ that determine a finite **intensity** $\lambda_k(t \mid x_{[0,t)}) \geq 0$ for each event type $k$ at each time $t > 0$ such that $p_k(t \mid x_{[0,t)}) = \lambda_k(t \mid x_{[0,t)})dt$. Then the log-likelihood of a temporal point process given the entire event sequence $x_{[0,T]}$ is

$$\sum_{i=1}^{I} \log \lambda_{k_i}(t_i \mid x_{[0,t_i)}) - \int_{t=0}^{T} \sum_{k=1}^{K} \lambda_k(t \mid x_{[0,t)})dt \tag{1}$$

Popular examples of temporal point processes include Poisson processes (Daley & Vere-Jones, 2007) as well as Hawkes processes (Hawkes, 1971) and their modern neural versions (Du et al., 2016; Mei & Eisner, 2017; Zuo et al., 2020; Zhang et al., 2020; Yang et al., 2022).

### 2.2 Task and Challenge: Long-Horizon Prediction and Cascading Errors

We are interested in predicting the future events over an *extended* time interval $(T, T']$. We call this task **long-horizon prediction** as the boundary $T'$ is so large that (with a high probability) many events will happen over $(T, T']$. A principled way to solve this task works as follows: we draw many possible future event sequences over the interval $(T, T']$, and then use this empirical distribution to answer questions such as "how many events of type $k = 3$ will happen over that interval".

A serious technical issue arises when we draw each possible future sequence. To draw an event sequence from an autoregressive model, we have to repeatedly draw the next event, append it to the history, and then continue to draw the next event conditioned on the new history. This process is prone to **cascading errors**: any error in a drawn event is likely to cause all the subsequent draws to differ from what they should be, and such errors will accumulate.

### 2.3 Globally Normalized Models: Hope and Difficulties

An ideal fix of this issue is to develop a **globally normalized model** for event sequences. For any time interval $[0, T]$, such a model will give a probability distribution that is normalized over all the possible *full sequences* on $[0, T]$ rather than over all the possible instantaneous subsequences within each $(t, t + dt)$. Technically, a globally normalized model assigns to each sequence $x_{[0,T]}$ a

score $\exp\left(-E(x_{[0,T]})\right)$ where $E$ is called **energy function**; the normalized probability of $x_{[0,T]}$ is proportional to its score: i.e., $p(x_{[0,T]}) \propto \exp\left(-E(x_{[0,T]})\right)$.

Had we trained a globally normalized model, we wish to enumerate all the possible $x_{[T,T']}$ for a given $x_{[0,T]}$ and select those which give the highest model probabilities $p(x_{[0,T']})$. Prediction made this way would not suffer cascading errors: the entire $x_{[0,T']}$ was jointly selected and thus the overall compatibility between the events had been considered.

However, training such a globally normalized probabilistic model involves computing the normalizing constant $\sum \exp\left(-E(x_{[0,T]})\right)$ where the summation $\sum$ is taken over all the possible sequences; it is intractable since there are *infinitely* many sequences. What's worse, it is also intractable to exactly sample from such a model; approximate sampling is tractable but expensive.

# 3 HYPRO: A Hybridly Normalized Neural Probabilistic Model

We propose HYPRO, a hybridly normalized neural probabilistic model that combines a temporal point process and an energy function: it enjoys both the efficiency of autoregressive models and the capacity of globally normalized models. Our model normalizes over (sub)sequences: for any given interval $[0, T]$ and its extension $(T, T')$ of interest, the model probability of the sequence $x_{(T,T')}$ is

$$p_{\text{HYPRO}}\left(x_{(T,T')} \mid x_{[0,T]}\right) = p_{\text{auto}}\left(x_{(T,T')} \mid x_{[0,T]}\right) \frac{\exp\left(-E_\theta(x_{[0,T']})\right)}{Z_\theta\left(x_{[0,T]}\right)} \tag{2}$$

where $p_{\text{auto}}$ is the probability under the chosen temporal point process and $E_\theta$ is an energy function with parameters $\theta$. The normalizing constant sums over all the possible continuations $x_{(T,T')}$ for a given prefix $x_{[0,T]}$: $Z_\theta\left(x_{[0,T]}\right) \overset{\text{def}}{=} \sum_{x_{(T,T')}} p_{\text{auto}}\left(x_{(T,T')} \mid x_{[0,T]}\right) \exp\left(-E_\theta(x_{[0,T']})\right)$.

The key advantage of our model over autoregressive models is that: the energy function $E_\theta$ is able to pick up the global features that may have been missed by the autoregressive base model $p_{\text{auto}}$; intuitively, the energy function fits the residuals that are not captured by the autoregressive model.

Our model is general: in principle, $p_{\text{auto}}$ can be any autoregressive model including those mentioned in section 2.1 and $E_\theta$ can be any function that is able to encode an event sequence to a real number. In section 5, we will introduce a couple of specific $p_{\text{auto}}$ and $E_\theta$ and experiment with them.

In this section, we focus on the training method and inference algorithm.

## 3.1 Training Objectives

Training our full model $p_{\text{HYPRO}}$ is to learn the parameters of the autoregressive model $p_{\text{auto}}$ as well as those of the energy function $E_\theta$. Maximum likelihood estimation (MLE) is undesirable: the objective would be $\log p_{\text{HYPRO}}\left(x_{(T,T')} \mid x_{[0,T]}\right) = \log p_{\text{auto}}\left(x_{(T,T')} \mid x_{[0,T]}\right) - E_\theta(x_{[0,T']}) - \log Z_\theta\left(x_{[0,T]}\right)$ where the normalizing constant $Z_\theta\left(x_{[0,T]}\right)$ is known to be *uncomputable* and *inapproximable* for a large variety of reasonably expressive functions $E_\theta$ (Lin & McCarthy, 2022).

We propose a training method that works around this normalizing constant. We first train $p_{\text{auto}}$ just like how previous work trained temporal point processes.[1] Then we use the trained $p_{\text{auto}}$ as a noise distribution and learn the parameters $\theta$ of $E_\theta$ by noise-contrastive estimation (NCE). Precisely, we sample $N$ noise sequences $x_{[T,T']}^{(1)}, \ldots, x_{[T,T']}^{(N)}$, compute the "energy" $E_\theta(x_{[0,T']}^{(n)})$ for each *completed* sequence $x_{[0,T']}^{(n)}$, and then plug those energies into one of the following training objectives.

Note that all the completed sequences $x_{[0,T']}^{(n)}$ share the same observed prefix $x_{[0,T]}$.

**Binary-NCE Objective.** We train a binary classifier based on the energy function $E_\theta$ to discriminate the true event sequence—denoted as $x_{[0,T']}^{(0)}$—against the noise sequences by maximizing

$$J_{\text{binary}} = \log \sigma\left(-E_\theta(x_{[0,T']}^{(0)})\right) + \sum_{n=1}^{N} \log \sigma\left(E_\theta(x_{[0,T']}^{(n)})\right) \tag{3}$$

---

[1]It can be done by either maximum likelihood estimation or noise-contrastive estimation: for the former, read Daley & Vere-Jones (2007); for the latter, read Mei et al. (2020b) which also has an in-depth discussion about the theoretical connections between these two parameter estimation principles.

where $\sigma(u) = \frac{1}{1+\exp(-u)}$ is the sigmoid function. By maximizing this objective, we are essentially pushing our energy function $E_\theta$ such that the observed sequences have *low* energy but the noise sequences have *high* energy. As a result, the observed sequences will be *more probable* under our full model $p_{\mathrm{HYPRO}}$ while the noise sequences will be *less probable*: see equation (2).

Theoretical guarantees of general Binary-NCE can be found in Gutmann & Hyvärinen (2010). For general conditional probability models like ours, Binary-NCE implicitly assumes self-normalization (Mnih & Teh, 2012; Ma & Collins, 2018): i.e., $Z_\theta\left(x_{[0,T]}\right) = 1$ is satisfied.

This type of training objective has been used to train a hybridly normalized text generation model by Deng et al. (2020); see section 4 for more discussion about its relations with our work.

**Multi-NCE Objective.** Another option is to use Multi-NCE objective[2], which means we maximize

$$J_{\mathrm{multi}} = -E_\theta(x^{(0)}_{[0,T']}) - \log \sum_{n=0}^{N} \exp\left(-E_\theta(x^{(n)}_{[0,T']})\right) \tag{4}$$

By maximizing this objective, we are pushing our energy function $E_\theta$ such that each observed sequence has *relatively lower* energy than the noise sequences sharing the same observed prefix. In contrast, $J_{\mathrm{binary}}$ attempts to make energies *absolutely* low (for observed data) or high (for noise data) without considering whether they share prefixes. This effect is analyzed in Analysis-III of section 5.2.

This $J_{\mathrm{multi}}$ objective also enjoys better statistical properties than $J_{\mathrm{binary}}$ since it doesn't assume self-normalization: the normalizing constant $Z_\theta\left(x_{[0,T]}\right)$ is neatly cancelled out in its derivation; see Appendix A.1 for a full derivation of both Binary-NCE and Multi-NCE.

Theoretical guarantees of Multi-NCE for discrete-time models were established by Ma & Collins (2018); Mei et al. (2020b) generalized them to temporal point processes.

**Considering Distances Between Sequences.** Previous work (LeCun et al., 2006; Bakhtin et al., 2019) reported that energy functions may be better learned if the distances between samples are considered. This has inspired us to design a regularization term that enforces such consideration.

Suppose that we can measure a well-defined "distance" between the true sequence $x^{(0)}_{[0,T']}$ and any noise sequence $x^{(n)}_{[0,T']}$; we denote it as $d(n)$. We encourage the energy of each noise sequence to be higher than that of the observed sequence by a margin; that is, we propose the following regularization:

$$\Omega = \sum_{n=1}^{N} \max\left(0, \beta d(n) + E_\theta(x^{(0)}_{[0,T']}) - E_\theta(x^{(n)}_{[0,T']})\right) \tag{5}$$

where $\beta > 0$ is a hyperparameter that we tune on the held-out development data. With this regularization, the energies of the sequences with larger distances will be pulled farther apart: this will help discriminate not only between the observed sequence and the noise sequences, but also between the noise sequences themselves, thus making the energy function $E_\theta$ more informed.

This method is general so the distance $d$ can be any appropriately defined metric. In section 5, we will experiment with an optimal transport distance specifically designed for event sequences.

Note that the distance $d$ in the regularization may be the final test metric. In that case, our method is directly optimizing for the final evaluation score.

**Generating Noise Sequences.** Generating event sequences from an autoregressive temporal point process has been well-studied in previous literature. The standard way is to call the **thinning algorithm** (Lewis & Shedler, 1979; Liniger, 2009). The full recipe for our setting is in Algorithm 1.

### 3.2 Inference Algorithm

Inference involves drawing future sequences $x_{(T,T']}$ from the trained full model $p_{\mathrm{HYPRO}}$; due to the uncomputability of the normalizing constant $Z(x_{[0,T]})$, exact sampling is intractable.

We propose a **normalized importance sampling** method to approximately draw $x_{(T,T']}$ from $p_{\mathrm{HYPRO}}$; it is shown in Algorithm 2. We first use the trained $p_{\mathrm{auto}}$ to be our proposal distribution and call the thinning algorithm (Algorithm 1) to draw proposals $x^{\langle 1 \rangle}_{[T,T']}, \ldots, x^{\langle M \rangle}_{[T,T']}$. Then we

---

[2]It was named as Ranking-NCE by Ma & Collins (2018), but we think Multi-NCE is a more appropriate name since it constructs a multi-class classifier over one correct answer and multiple incorrect answers.

---

**Algorithm 1** Generating Noise Sequences.

---

**Input:** an event sequence $x_{[0,T]}$ over the given interval $[0,T]$ and an interval $(T,T')$ of interest; trained autoregressive model $p_{\text{auto}}$ and number of noise samples $N$
**Output:** a collection of noise sequences
1: **procedure** DRAWNOISE($x_{[0,T]}, T', p_{\text{auto}}, N$)
2:    **for** $n = 1$ **to** $N$ :
3:       ▷ *use the thinning algorithm to draw each noise sequences from the autoregressive model $p_{auto}$*
4:       ▷ *in particular, call the method in Algorithm 3 that is described in Appendix A.2*
5:       $x_{(T,T')}^{(n)} \leftarrow$ THINNING($x_{[0,T]}, T', p_{\text{auto}}$)
6:    **return** $x_{(T,T')}^{(1)}, \ldots, x_{(T,T')}^{(N)}$

---

reweight those proposals with the *normalized* weights $w^{\langle m \rangle}$ that are defined as

$$w^{\langle m \rangle} \stackrel{\text{def}}{=} \frac{p_{\text{HYPRO}}(x_{[T,T']}^{\langle m \rangle})/p_{\text{auto}}(x_{[T,T']}^{\langle m \rangle})}{\sum_{m'=1}^{M} p_{\text{HYPRO}}(x_{[T,T']}^{\langle m' \rangle})/p_{\text{auto}}(x_{[T,T']}^{\langle m' \rangle})} = \frac{\exp\left(-E_\theta(x_{[0,T']}^{\langle m \rangle})\right)}{\sum_{m'=1}^{M} \exp\left(-E_\theta(x_{[0,T']}^{\langle m' \rangle})\right)} \tag{6}$$

This collection of weighted proposals is used for the long-horizon prediction over the interval $(T,T']$: if we want the most probable sequence, we return the $x_{[T,T']}^{\langle m \rangle}$ with the largest weight $w^{\langle m \rangle}$; if we want a minimum Bayes risk prediction (for a specific risk metric), we can use existing methods (e.g., the consensus decoding method in Mei et al. (2019)) to compose those weighted samples into a single sequence that minimizes the risk. In our experiments (section 5), we used the most probable sequence.

Note that our sampling method is *biased* since the weights $w^{\langle m \rangle}$ are *normalized*. Unbiased sampling in our setting is intractable since that will need our weights to be unnormalized: i.e., $w \stackrel{\text{def}}{=} p_{\text{HYPRO}}/p_{\text{auto}} = \exp\left(-E_\theta\right)/Z$ which circles back to the problem of $Z$'s uncomputability. Experimental results in section 5 show that our method indeed works well in practice despite that it is biased.

---

**Algorithm 2** Normalized Importance Sampling for Long-Horizon Prediction.

---

**Input:** an event sequence $x_{[0,T]}$ over the given interval $[0,T]$ and an interval $(T,T')$ of interest; trained autoregressive model $p_{\text{auto}}$ and engergy function $E_\theta$, number of proposals $M$
**Output:** a collection of weighted proposals
1: **procedure** NIS($x_{[0,T]}, T', p_{\text{auto}}, E_\theta, M$)
2:    ▷ *use normalized importance sampling to approximately draw M proposals from $p_{HYPRO}$*
3:    $x_{[T,T']}^{\langle 1 \rangle}, \ldots, x_{[T,T']}^{\langle M \rangle} \leftarrow$ DRAWNOISE($x_{[0,T]}, T', p_{\text{auto}}, M$)       ▷ *see Algorithm 1*
4:    construct completed sequences $x_{[0,T']}^{\langle 1 \rangle}, \ldots, x_{[0,T']}^{\langle M \rangle}$ by appending each $x_{[T,T']}^{\langle m \rangle}$ to $x_{[0,T]}$
5:    compute the exponential of minus energy $e^{\langle m \rangle} = \exp\left(-E_\theta(x_{[0,T']}^{\langle m \rangle})\right)$ for each proposal
6:    compute the normalized weights $w^{\langle m \rangle} = e^{\langle m \rangle} / \sum_{m'=1}^{M} e^{\langle m' \rangle}$
7:    **return** $(w^{\langle 1 \rangle}, x_{[T,T']}^{\langle 1 \rangle}), \ldots, (w^{\langle M \rangle}, x_{[T,T']}^{\langle M \rangle})$       ▷ *return the collection of weighted proposals*

---

## 4 Related work

Over the recent years, various neural temporal point processes have been proposed. Many of them are built on recurrent neural networks, or LSTMs (Hochreiter & Schmidhuber, 1997); they include Du et al. (2016); Mei & Eisner (2017); Xiao et al. (2017a,b); Omi et al. (2019); Shchur et al. (2020); Mei et al. (2020a); Boyd et al. (2020). Some others use Transformer architectures (Vaswani et al., 2017; Radford et al., 2019): in particular, Zuo et al. (2020); Zhang et al. (2020); Enguehard et al. (2020); Sharma et al. (2021); Zhu et al. (2021); Yang et al. (2022). All these models all autoregressive: they define the probability distribution over event sequences in terms of a sequence of locally-normalized conditional distributions over events given their histories.

Energy-based models, which have a long history in machine learning (Hopfield, 1982; Hinton, 2002; LeCun et al., 2006; Ranzato et al., 2007; Ngiam et al., 2011; Xie et al., 2019), define the distribution over sequences in a different way: they use energy functions to summarize each possible sequence into a scalar (called energy) and define the unnormalized probability of each sequence in terms of its energy, then the probability distribution is normalized across all sequences; thus, they are also

called globally normalized models. Globally normalized models are a strict generalization of locally normalized models (Lin et al., 2021): all the locally normalized models are globally normalized; but the converse is not true. Moreover, energy functions are good at capturing global features and structures (Pang et al., 2021; Du & Mordatch, 2019; Brakel et al., 2013). However, the normalizing constants of globally normalized models are often uncomputable (Lin & McCarthy, 2022). Existing work that is most similar to ours is the energy-based text generation models of Bakhtin et al. (2019) and Deng et al. (2020) that train energy functions to reweight the outputs generated by pretrained autoregressive models. The differences are: we work on different kinds of sequential data (continuous-time event sequences vs. discrete-time natural language sentences), and thus the architectures of our autoregressive model and energy function are different from theirs; additionally, we explored a wider range of training objectives (e.g., Multi-NCE) than they did.

The task of long-horizon prediction has drawn much attention in several machine learning areas such as regular time series analysis (Yu et al., 2019; Le Guen & Thome, 2019), natural language processing (Guo et al., 2018; Guan et al., 2021), and speech modeling (Oord et al., 2016). Deshpande et al. (2021) is the best-performing to-date in long-horizon prediction of event sequences: they adopt a hierarchical architecture similar to ours and use a ranking objective based on the counts of the events. Their method can be regarded as a special case of our framework (if we let our energy function read the counts of the events), and our method works better in practice (see section 5).

# 5 Experiments

We implemented our methods with PyTorch (Paszke et al., 2017). Our code can be found at `https://github.com/alipay/hypro_tpp` and `https://github.com/iLampard/hypro_tpp`. Implementation details can be found in Appendix B.2.

## 5.1 Experimental Setup

Given a train set of sequences, we use the full sequences to train the autoregressive model $p_{\text{auto}}$ by maximizing equation (1). To train the energy function $E_\theta$, we need to split each sequence into a prefix $x_{[0,T]}$ and a continuation $x_{(T,T']}$: we choose $T$ and $T'$ such that there are 20 event tokens within $(T, T']$ on average. During testing, for each prefix $x_{[0,T]}$, we draw 20 weighted samples (Algorithm 2) and choose the highest-weighted one as our prediction $\hat{x}_{(T,T']}$. We evaluate our predictions by:

- The root of mean square error (RMSE) of the number of the tokens of each event type: for each type $k$, we count the number of type-$k$ tokens in the true continuation—denoted as $C_k$—as well as

  that in the prediction—denoted as $\hat{C}_k$; then the mean square error is $\sqrt{\frac{1}{K} \sum_{k=1}^{K} \left( C_k - \hat{C}_k \right)^2}$.

- The optimal transport distance (OTD) between event sequences defined by Mei et al. (2019): for any given prefix $x_{[0,T]}$, the distance is defined as the minimal cost of editing the prediction $\hat{x}_{(T,T']}$ (by inserting or deleting events, changing their occurrence times, and changing their types) such that it becomes exactly the same as the true continuation $x_{(T,T']}$.

We did experiments on two real-world datasets (see Appendix B.1 for dataset details):

- **Taobao** (Alibaba, 2018)**.** This public dataset was created and released for the 2018 Tianchi Big Data Competition. It contains time-stamped behavior records (e.g., browsing, purchasing) of anonymized users on the online shopping platform Taobao from November 25 through December 03, 2017. Each category group (e.g., men's clothing) is an event type, and we have $K = 17$ event types. We use the browsing sequences of the most active 2000 users; each user has a sequence. Then we randomly sampled disjoint train, dev and test sets with 1300, 200 and 500 sequences. The time unit is 3 hours; the average inter-arrival time is 0.06 (i.e., 0.18 hour), and we choose the prediction horizon $T' - T$ to be 1.5 that approximately covers 20 event tokens.

- **Taxi** (Whong, 2014)**.** This dataset tracks the time-stamped taxi pick-up and drop-off events across the five boroughs of the New York city; each (borough, pick-up or drop-off) combination defines an event type, so there are $K = 10$ event types in total. We work on a randomly sampled subset of 2000 drivers and each driver has a sequence. We randomly sampled disjoint train, dev and test sets with 1400, 200 and 400 sequences. The time unit is 1 hour; the average inter-arrival time is 0.22, and we set the prediction horizon to be 4.5 that approximately covers 20 event tokens.

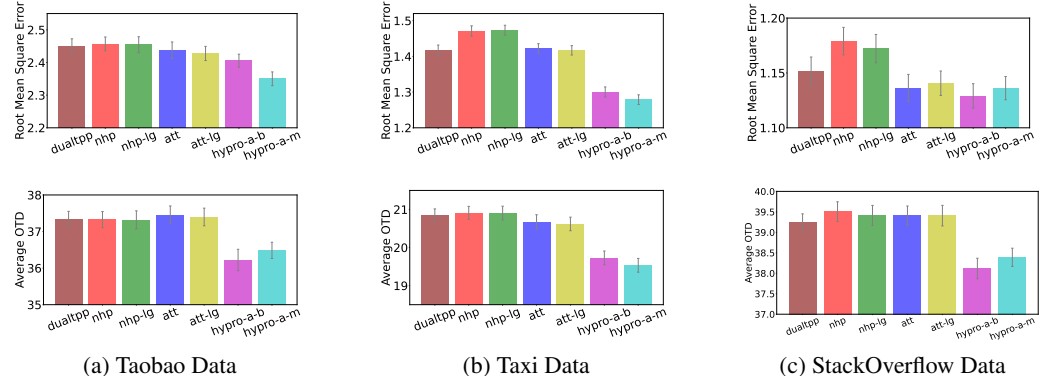

Figure 1: Performance of all the methods on Taobao (1a), Taxi (1b) and StackOverflow (1c) datasets, measured by RMSE (up) and OTD (down). In each figure, the models from left to right are: DualTPP (dualtpp); NHP (nhp); NHP with more parameters (nhp-lg); AttNHP (att); AttNHP with more parameters (att-lg); our HYPRO with Transformer energy function trained via Binary-NCE (hypro-a-b) and Multi-NCE (hypro-a-m).

- **StackOverflow** (Leskovec & Krevl, 2014). This dataset has two years of user awards on a question-answering website: each user received a sequence of badges and there are $K = 22$ different kinds of badges in total. We randomly sampled disjoint train, dev and test sets with $1400, 400$ and $400$ sequences from the dataset. The time unit is $11$ days; the average inter-arrival time is $0.95$ and we set the prediction horizon to be $20$ that approximately covers 20 event tokens.

We choose two strong autoregressive models as our base model $p_{\text{auto}}$:

- **Neural Hawkes process (NHP)** (Mei & Eisner, 2017). It is an LSTM-based autoregressive model that has demonstrated effective at modeling event sequences in various domains.

- **Attentative neural Hawkes process (AttNHP)** (Yang et al., 2022). It is an attention-based autoregressive model—like Transformer language model (Vaswani et al., 2017; Radford et al., 2019)—whose performance is comparable to or better than that of the NHP as well as other attention-based models (Zuo et al., 2020; Zhang et al., 2020).

For the energy function $E_\theta$, we adapt the **continuous-time Transformer module** of the AttNHP model: the Transformer module embeds the given sequence of events $x$ into a fixed-dimensional vector (see section-2 of Yang et al. (2022) for details), which is then mapped to a scalar $\in \mathbb{R}$ via a multi-layer perceptron (MLP); that scalar is the energy value $E_\theta(x)$.

We first train the two base models NHP and AttNHP; they are also used as the baseline methods that we will compare to. To speed up energy function training, we use the pretrained weights of the AttNHP to initialize the Transformer part of the energy function; this trick was also used in Deng et al. (2020) to bootstrap the energy functions for text generation models. As the full model $p_{\text{HYPRO}}$ has significantly more parameters than the base model $p_{\text{auto}}$, we also trained larger NHP and AttNHP with comparable amounts of parameters as extra baselines. Additionally, we also compare to the **DualTPP** model of Deshpande et al. (2021). Details about model parameters are in Table 2 of Appendix B.3.

### 5.2 Results and Analysis

The main results are shown in Figure 1. The OTD depends on the hyperparameter $C_{\text{del}}$, which is the cost of deleting or adding an event token of any type, so we used a range of values of $C_{\text{del}}$ and report the averaged OTD in Figure 1; OTD for each specific $C_{\text{del}}$ can be found in Appendix B.4. As we can see, NHP and AttNHP work the worst in most cases. DualTPP doesn't seem to outperform these autoregressive baselines even though it learns to rerank sequences based on their macro statistics; we believe that it is because DualTPP's base autoregressive model is not as powerful as the state-of-the-art NHP and AttNHP and using macro statistics doesn't help enough. Our HYPRO method works significantly better than these baselines.

**Analysis-I: Does the Cascading Error Exist?** Handling cascading errors is a key motivation for our framework. On the Taobao dataset, we empirically confirmed that this issue indeed exists.

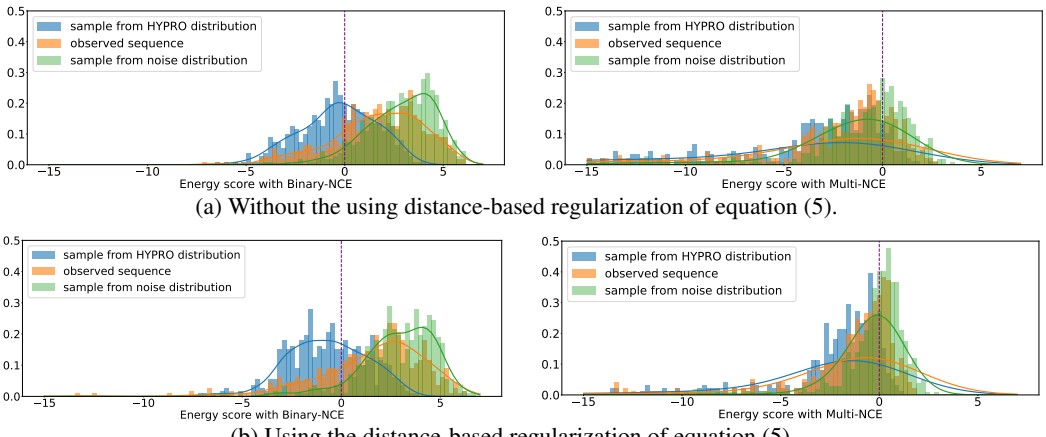

(a) Without the using distance-based regularization of equation (5).

(b) Using the distance-based regularization of equation (5).

Figure 2: Energy scores computed on the held-out development data of Taobao dataset.

We first investigate whether the event type prediction errors are cascaded through the subsequent events. We grouped the sequences based on how early the first event type prediction error was made and then compared the event type prediction error rate on the subsequent events:

- when the first error is made on the first event token, the base AttNHP model has a 71.28% error rate on the subsequent events and our hybrid model has a much lower 66.99% error rate.

- when the first error is made on the fifth event token, the base AttNHP model has a 58.88% error rate on the subsequent events and our hybrid model has a much lower 51.89% error rate.

- when the first error is made on the tenth event token, the base AttNHP model has a 44.48% error rate on the subsequent events and our hybrid model has a much lower 35.58% error rate.

Obviously, when mistakes are made earlier in the sequences, we tend to end up with a higher error rate on the subsequent predictions; that means event type prediction errors are indeed cascaded through the subsequent predictions. Moreover, in each group, our hybrid model enjoys a lower prediction error; that means it indeed helps mitigate this issue.

We then investigate whether the event time prediction errors are cascaded. For this, we performed a linear regression: the independent variable $x$ is the absolute error of the prediction on the time of the first event token; the dependent variable $y$ is the averaged absolute error of the prediction on the time of the subsequent event tokens. Our fitted linear model is $y = 0.7965x + 0.3219$ where the p-value of the coefficient of $x$ is $\approx 0.0001 < 0.01$. It means that the time prediction errors are also cascaded through the subsequent predictions.

**Analysis-II: Energy Function or Just More Parameters?** The larger NHP and AttNHP have almost the same numbers of parameters with HYPRO, but their performance is only comparable to the smaller NHP and AttNHP. That is to say, simply increasing the number of parameters in an autoregressive model will not achieve the performance of using an energy function.

To further verify the usefulness of the energy function, we also compared our method with another baseline method that ranks the completed sequences based on their probabilities under the base model, from which the continuations were drawn. This baseline is similar to our proposed HYPRO framework but its scorer is the base model itself. In our experiments, this baseline method is not better than our method; details can be found in Appendix B.5.

Overall, we can conclude that the energy function $E_\theta$ is essential to the success of HYPRO.

**Analysis-III: Binary-NCE vs. Multi-NCE.** Both Binary-NCE and Multi-NCE objectives aim to match our full model distribution $p_{\text{HYPRO}}$ with the true data distribution, but Multi-NCE enjoys better statistical properties (see section 3) and achieved better performance in our experiments (see Figure 1). In Figure 2a, we display the distributions of the energy scores of the observed sequences, $p_{\text{HYPRO}}$-generated sequences, and $p_{\text{auto}}$-generated noise sequences: as we can see, the distribution of $p_{\text{HYPRO}}$ is different from that of noise data and indeed closer to that of real data.

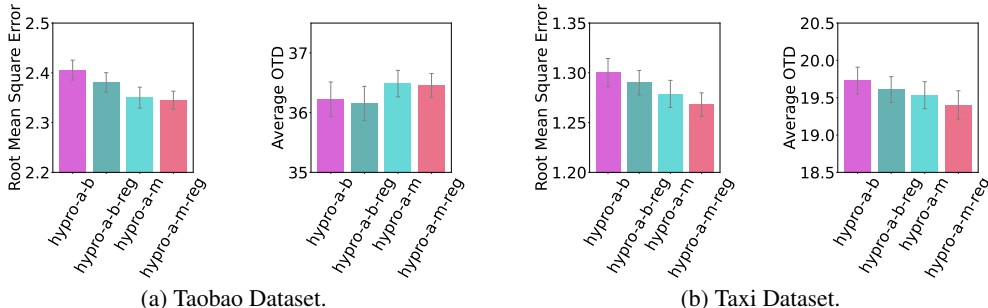

(a) Taobao Dataset.

(b) Taxi Dataset.

Figure 3: Adding the regularization term $\Omega$. In each figure, the suffix -reg denotes "with regularization".

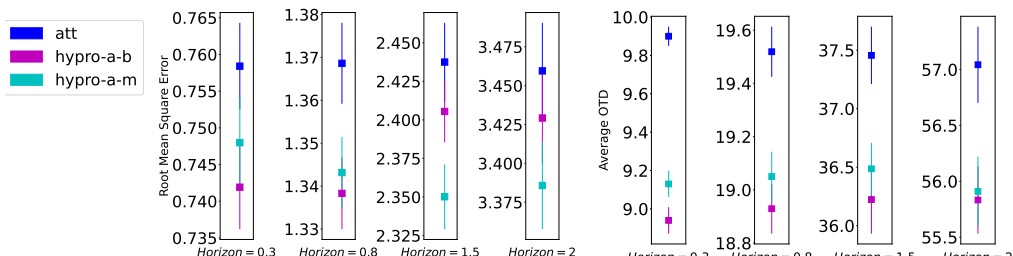

Figure 4: HYPRO vs. AttNHP for different horizons on Taobao Dataset.

**Analysis-IV: Effects of the Distance-Based Regularization $\Omega$.** We experimented with the proposed distance-based regularization $\Omega$ in equation (5); as shown in Figure 3, it slightly improves both Binary-NCE and Multi-NCE. As shown in Figure 2b, the regularization makes a larger difference in the Binary-NCE case: the energies of $p_{\text{HYPRO}}$-generated sequences are pushed further to the left.

We did the paired permutation test to verify the statistical significance of our regularization technique; see Appendix B.6 for details. Overall, we found that the performance improvements of using the regularization are strongly significant in the Binary-NCE case (p-value $< 0.05$ ) but not significant in the Multi-NCE case (p-value $\approx 0.1$). This finding is consistent with the observations in Figure 2.

**Analysis-V: Effects of Prediction Horizon.** Figure 4 shows how well our method performs for different prediction horizons. On Taobao Dataset, we experimented with horizon being $0.3, 0.8, 1.5, 2$, corresponding to approximately $5, 10, 20, 30$ event tokens, and found that our HYPRO method improves significantly and consistently over the autoregressive baseline AttNHP.

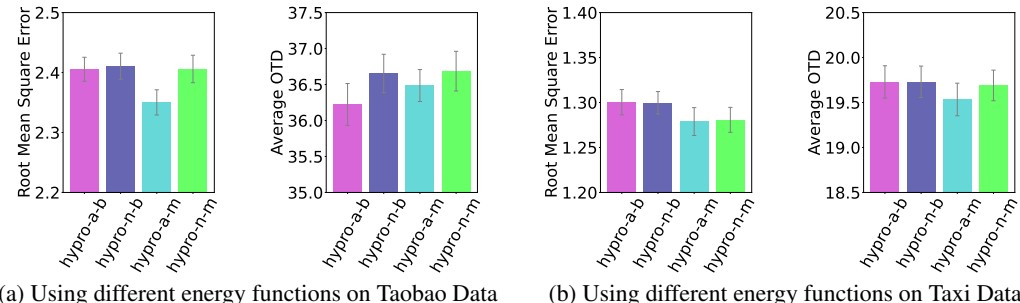

(a) Using different energy functions on Taobao Data

(b) Using different energy functions on Taxi Data

Figure 5: Using different energy functions. In each figure, the suffix -n-b and -n-m denote using a continuous-time LSTM as the energy function trained by Binary-NCE and Multi-NCE, respectively.

**Analysis-VI: Different Energy Functions.** So far we have only shown the results and analysis of using the continuous-time Transformer architecture as the energy function $E_\theta$. We also experimented with using a continuous-time LSTM (Mei & Eisner, 2017) as the energy function and found that it never outperformed the Transformer energy function in our experiments; see Figure 5. We think

it is because the Transformer architectures are better at embedding contextual information than LSTMs (Vaswani et al., 2017; Pérez et al., 2019; O'Connor & Andreas, 2021).

**Analysis-VII: Negative Samples.** On the Taobao dataset, we analyzed how the number of negative samples affects training and inference. We experimented with the Binary-NCE objective without the distance regularization (i.e., hypro-a-b). The results are in Figure 6. During training, increasing the number of negative samples from 1 to 5 has brought improvements but further increasing it to 10 does not. During inference, the results are improved when we increase the number of samples from 5 to 20, but they stop improving when we further increase it. Throughout the paper, we used 5 in training and 20 in inference (Appendix B.3).

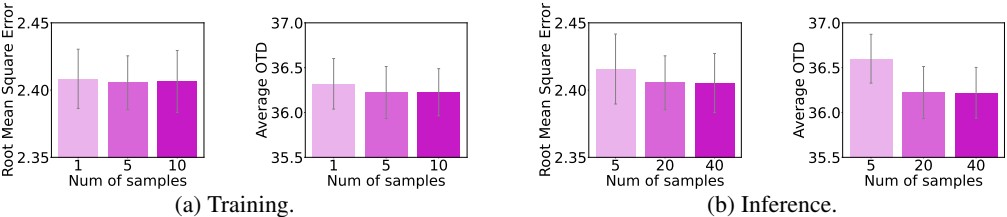

Figure 6: Effects of the number of negative samples in training and inference on the Taobao dataset.

# 6 Conclusion

We presented HYPRO, a hybridly normalized neural probabilistic model for the task of long-horizon prediction of event sequences. Our model consists of an autoregressive base model and an energy function: the latter learns to reweight the sequences drawn from the former such that the sequences that appear more realistic as a whole can end up with higher probabilities under our full model. We developed two training objectives that can train our model without computing its normalizing constant, together with an efficient inference algorithm based on normalized importance sampling. Empirically, our method outperformed current state-of-the-art autoregressive models as well as a recent non-autoregressive model designed specifically for the same task.

# 7 Limitations and Societal Impacts

**Limitations.** Our method uses neural networks, which are typically data-hungry. Although it worked well in our experiments, it might still suffer compared to non-neural models if starved of data. Additionally, our method requires the training sequences to be sufficiently long so it can learn to make long-horizon predictions; it may suffer if the training sequences are short.

**Societal Impacts.** Our paper develops a novel probabilistic model for long-horizon prediction of event sequences. By describing the model and releasing code, we hope to facilitate probabilistic modeling of continuous-time sequential data in many domains. However, like many other machine learning models, our model may be applied to unethical ends. For example, its abilities of better fitting data and making more accurate predictions could potentially be used for unwanted tracking of individual behavior, e.g. for surveillance.

# Acknowledgments

This work was supported by a research gift to the last author by Adobe Research. We thank the anonymous NeurIPS reviewers and meta-reviewer for their constructive feedback. We also thank our colleagues at Ant Group for helpful discussion.

