# OpenReview forum: "HYPRO: A Hybridly Normalized Probabilistic Model for Long-Horizon Prediction of Event Sequences"
_NeurIPS.cc/2022/Conference — NeurIPS 2022 Accept_

### Official Review · Reviewer_urPE · 2022-06-22

**Rating:** 7
**Confidence:** 4
**Soundness:** 3 good
**Presentation:** 2 fair
**Contribution:** 2 fair

**Summary:**

The paper proposes a hybrid approach using an autoregressive + energy based model to model event sequences. Prior approaches model event sequences data using just an autoregressive model. This paper trains an energy-based model to rank the samples from the autoregressive model. The sample with the highest weight across a number of samples is then used as the model output.

2 different objective functions (Binary-NCE + Multi-NCE) are used to train the energy-based model. These functions assign lower energy to ground-truth sequences and higher energy to noise. An additional regularization that enforces a margin based on “edit distance” between the real and model outputs are also explored. The authors outperform prior works (NHP and AttNHP) with 2x parameters and DUALTPP on the Taobao and Taxi dataset on the RMSE and OTD metrics.


**Questions:**

Major Questions:
* [Experiments:] In L207 and L 209, the authors train the transformer model on 1300 and 1440 sequences on the Taobao and Taxi dataset. Transformers are generally known to be data hungry, so this is a bit surprising that the model works well on such a small dataset. What are the tricks (if any) that the authors employ to get good results on such a small dataset?
* [Experiments:] It is a bit intriguing that the 2x larger models in Figure 2a and Figure 2b get the same results as the smaller baselines. Do the authors have intuition on this? Does the performance on the training set saturate as well?
* [Experiments:] Instead of training a separate energy based model to assign weights to the samples in Eq 6), a simple baseline is to assign weights using the likelihood of the autoregressive model itself. What are the results using this?
* [Claims:] I appreciate that the authors are transparent in L121 and L183-L184 that similar approaches have been used in text-generation. I suggest that the authors be more upfront about their contributions in this paper. For example, the technical innovations in the introduction from L31 - L44 imply that this paper is the first to propose the hybrid model. In my opinion, It should be rephrased to say that this paper is the first to adapt the hybrid model to event sequence data with XYZ architectural + training modifications, since the hybrid model itself is not the main contribution of the paper.
* [Claims:] L183 says that "the architectures of our autoregressive model and energy function are different from theirs;". What exactly are the architectural changes? What changes were made to get this model to work on event sequence data?
* [Claims:] Is the importance sampling method the same as Deng et al or were changes made to this to adapt this model to event sequence data as well?

Some more details are required to make the draft self-contained.
* What exactly is the output distribution? Is $\lambda_k$ a Gaussian distribution per k?
* How is the 2nd term optimized in practice? Is the integral over time approximated with uniform sampling or a discrete grid?
* The paper says from L73 - L78 that energy-based models give a probability normalized across all sequences. But so do, autoregressive models, since the exact density at a point is computable. Can the authors please clarify this? In that case, the motivation from L72 - L78 might be a bit handwavy.
* How is the upper bound in Line 7 of Algorithm 1 computed in practice since the domain of $t$ is infinite, i.e $t \in (t_0, \inf)$?

Not crucial but Good to have/know:
* [Experiments:] The authors perform the experiments with 20 samples. Do the results get better if the number of samples are increased?
* [Experiments:] The regularization term explicitly incorporates the edit distances in the loss. So the Figure 3a) and 3b) Right could be misleading since the test metric is used in optimization.
* [Experiments:] The BinaryNCE objective has one positive sample and N negative samples per gradient update. What is the effect of this class imbalance?
* If I understand correctly, the entire framework can be viewed as a GAN with the autoregressive model as the generator and the energy based model as a discriminator and a single round of updates. Do the authors have thoughts on this?
* Should Eq2) be proportional to rather than equal? Even though the energy model and autoregressive model independenly output probabilities (that are normalized across all sequences), their product might not.

Minor:
* I think there is an Integral missing at L56.

**Limitations:**

Yes

**Strengths And Weaknesses:**

The entire paper is well written and easy to read. The empirical results are extremely strong.

I have a few questions on the experiments, claims and the draft (see below). I am currently borderline accept because of the strong emprical results. I am happy to adjust my rating if the authors can answer the makor questions below convincingly

---

> ### Author Response · Authors · 2022-08-02
> **point-by-point reply 4**
>
> > [Experiments:] The authors perform the experiments with 20 samples. Do the results get better if the number of samples are increased?
>
> Please see [Ablation: Number of Samples in Eval].
>
> > [Experiments:] The regularization term explicitly incorporates the edit distances in the loss. So the Figure 3a) and 3b) Right could be misleading since the test metric is used in optimization.
>
> You are right: it is in the optimization. We will surely clarify this in the final version.
> However, involving the test metric in the training optimization is also a principled way to do risk-minimization: e.g. risk-minimization training techniques proposed in the NLP community (translation and generation) involve their final eval scores (e.g., BLUE, ROGUE) in their training optimization. E.g.,
> Minimum Risk Training for Neural Machine Translation. Shen et al. ACL 2016.
>
> > [Experiments:] The BinaryNCE objective has one positive sample and N negative samples per gradient update. What is the effect of this class imbalance?
>
> We want N to be large enough so it provides useful training signals. But we do not want it to be too large since we do not want it to dominate the objective.
> In practice, N can be tuned as a hyperparameter.
> Please see [Ablation: Number of Negative Samples in Training].
>
> > If I understand correctly, the entire framework can be viewed as a GAN with the autoregressive model as the generator and the energy based model as a discriminator and a single round of updates. Do the authors have thoughts on this?
>
> You are exactly right: the energy function is the discriminator and the autoregressive model is the generator. But a subtle difference is: in GAN, one usually aims to train the generator and the discriminator is auxiliary; in our setting, we aim to train the discriminator (i.e., energy function) and the discriminator assists with that. We do want the autoregressive model to be decent such that the discrimination task would be difficult---that way, we will end up learning a good discriminator.
>
> In general, NCE framework is a lot like GAN: this NCE-GAN relation has been discussed in
> Noise-Contrastive Estimation for Multivariate Point Processes, Mei et al, NeurIPS, 2020.
>
> > Should Eq2) be proportional to rather than equal? Even though the energy model and autoregressive model independently output probabilities (that are normalized across all sequences), their product might not.
>
> It is a great catch. Thank you.
> Yes, you are right. Or we can redefine the partition function Z to be the sum of $p_{\text{auto}} \exp(-E)$.
>
> > I think there is an Integral missing at L56.
>
> For infinitesimal dt, having the integral or not is the same since $\lambda_k(t)$ can be regarded to be constant over the infinitesimal interval $[t,t+dt)$.
>
> To be precise, you are right that the infinitesimal probability at time $t$ should be
> $\int_{t}^{t+dt} \lambda_k(s) ds$ where $dt$ is infinitesimal.
> But please note that $\lambda_k(s)$ can be regarded to be the same as $\lambda_k(t)$ for $s \in [t, t+dt)$.
> Therefore, the integral formula will reduce to $\lambda_k(t) dt$, which is what you see at line-56.

---

> > ### Comment · Reviewer_urPE · 2022-08-08
> > **Rebuttal Response**
> >
> > I thank the authors for their comprehensive rebuttal and the new experiments.
> >
> > For the final version, I strongly encourage and expect the authors to restructure the introduction to provide more emphasis on their precise technical contributions as agreed upon.
> >
> > Since the thinning algorithm for sampling is not a contribution of this paper, the authors can consider moving this to the appendix. In the space left, the authors can add the additional experiments from the rebuttal, i.e the StackOverflow dataset and some of the ablations.

---

> > > ### Author Response · Authors · 2022-08-08
> > > **Thank you. The final version will be structured as we agreed.**
> > >
> > > Thank you for your supportive feedback.
> > >
> > > We will surely structure the final version as we have discussed and agreed.
> > > We will also include all the new results in the final version.

---

> ### Author Response · Authors · 2022-08-02
> **point-by-point reply 3**
>
> > The paper says from L73 - L78 that energy-based models give a probability normalized across all sequences. But so do, autoregressive models, since the exact density at a point is computable. Can the authors please clarify this? In that case, the motivation from L72 - L78 might be a bit handwavy.
>
> You are right that both autoregressive models and energy-based models define proper distributions. However, their normalization is different.
>
> Autoregressive models use local normalization: it normalizes at each time $t$ and the normalization at $t$ doesn't consider anything at times $> t$.
> Energy-based models use global normalization: they do not normalize at each time $t$; the normalization is taken over (infinitely many) all the sequences of interval $[0, T]$.
>
> Therefore, as you may have noticed: all locally normalized models are also globally normalized; not all globally normalized models are locally normalized. In other words, global normalization is a weaker assumption than local normalization---that is why a globally-normalized model tends to be more expressive. We will clarify this in our final version.
>
> > How is the upper bound in Line 7 of Algorithm 1 computed in practice since the domain of  is infinite, i.e ?
>
> Mei et al. 2017 and Yang et al. 2022 described ways to compute the upper bounds for NHP and AttNHP, respectively. We used their methods. We will clarify this in the paper.
> The Neural Hawkes Process: A Neurally Self-Modulating Multivariate Point Process, Mei et al, NeurIPS, 2017.
> Transformer Embeddings of Irregularly Spaced Events and Their Participants, Yang et al, ICLR 2022.

---

> ### Author Response · Authors · 2022-08-02
> **point-by-point reply 2**
>
> > [Claims:] L183 says that "the architectures of our autoregressive model and energy function are different from theirs;". What exactly are the architectural changes? What changes were made to get this model to work on event sequence data?
>
> We used continuous-time LSTM and continuous-time Transformer to handle time-stamped event data. The choice of $p_{\text{auto}}$ and $E$ are explained in line-223 to line-233.
>
> > [Claims:] Is the importance sampling method the same as Deng et al or were changes made to this to adapt this model to event sequence data as well?
>
> We both use normalized importance sampling. But we return the collection of samples with weights while they use a resampling step to return only one sample.
> Our returned collection could facilitate the possible step of MBR decoding (line-158). (Though we didn't experiment with it in this paper since it is not within the main scope.)
> Moreover, our sampling method has to draw event sequences, which is more sophisticated than drawing text sequences and has to call the thinning algorithm (Alg-1: ln-4 to ln-14).
>
> > Some more details are required to make the draft self-contained.
> > What exactly is the output distribution? Is a Gaussian distribution per k?
>
> No. $\lambda_k(t)$ is the intensity function and it is introduced and discussed in Section 2.1. For a more in-depth discussion of intensities, one could read the seminal textbook:
> Daley, D. J. and Vere-Jones, D. An Introduction to the Theory of Point Processes, Volume II: General Theory and Structure. Springer, 2007.
>
> Particularly, for a given $k$ $t$ and history, the intensity $\lambda_k(t)$ is deterministic (thus not following a Gaussian). What's stochastic is the event occurrence at any time $t$: the infinitesimal probability of an event token of $k$ occuring at time $t$ is exactly $\lambda_k(t) dt$. Please see Section 2.1 for more discussion.
>
> > How is the 2nd term optimized in practice? Is the integral over time approximated with uniform sampling or a discrete grid?
>
> It is approximated with Monte Carlo sampling. Time points are uniformly sampled from $(0, T)$. The estimate is unbiased. This trick was used in previous literature.
>
> Note that our proposed training objectives (eq-3,4,5) doesn't have this integral so it doesn't need such MC sampling.

---

> ### Author Response · Authors · 2022-08-02
> **point-by-point reply 1**
>
> Thank you for your constructive feedback and being supportive. We appreciate it and have added new experiment results to resolve your concerns; see [New Results].
> In this message, we will try to answer your remaining questions.
>
> > [Experiments:] In L207 and L 209, the authors train the transformer model on 1300 and 1440 sequences on the Taobao and Taxi dataset. Transformers are generally known to be data hungry, so this is a bit surprising that the model works well on such a small dataset. What are the tricks (if any) that the authors employ to get good results on such a small dataset?
>
> We only had one GPU for our experiments, which restricted ourselves to using small data. We didn't do anything special except that we used small models (e.g., ~20K params).
>
> > [Experiments:] It is a bit intriguing that the 2x larger models in Figure 2a and Figure 2b get the same results as the smaller baselines. Do the authors have intuition on this? Does the performance on the training set saturate as well?
>
> We tuned hyperparameters, ran each experiment until it converged, and selected model weights based on performance on held-out dev data.
> Having more parameters does not make the autoregressive model work better: that actually gives a clean and sharp verification that the energy function is useful---HYPRO doesn't win by only having more params; it wins by using those params more wisely.
>
> > [Experiments:] Instead of training a separate energy based model to assign weights to the samples in Eq 6), a simple baseline is to assign weights using the likelihood of the autoregressive model itself. What are the results using this?
>
> It doesn't work better. Please see [New Results - New Baseline].
>
> > [Claims:] I appreciate that the authors are transparent in L121 and L183-L184 that similar approaches have been used in text-generation. I suggest that the authors be more upfront about their contributions in this paper. For example, the technical innovations in the introduction from L31 - L44 imply that this paper is the first to propose the hybrid model. In my opinion, It should be rephrased to say that this paper is the first to adapt the hybrid model to event sequence data with XYZ architectural + training modifications, since the hybrid model itself is not the main contribution of the paper.
>
> We will surely make this change. Please see [Technical Contribution] for a little more discussion.

---

### Official Review · Reviewer_XUS4 · 2022-07-10

**Rating:** 6
**Confidence:** 4
**Soundness:** 2 fair
**Presentation:** 4 excellent
**Contribution:** 2 fair

**Summary:**

The authors propose a two-stage generative model for event sequences, with an emphasis on event forecasting over (relatively) long horizons. The first stage is simply a standard autoregressive model for event data in continuous time. There is already a substantial literature on such models - both classical temporal point processes and their neural counterparts - but the authors argue that this autoregressive structure leads to compounding errors for long-horizon prediction. Therefore the authors propose a second stage that reweights the probability of an autoregressively generated sequence using an energy function intended to capture plausible *global* structure. The authors propose corresponding training and conditional generation methods for this model, and they compare its performance to some baselines on two multivariate event datasets.

**Questions:**

1. Can the authors provide some citations or evidence for compounding errors in long-horizon event forecasting?

2. The details of the NCE training of the energy function lead me to wonder whether the second stage of the HYPRO model is really just enforcing an after-the-fact regularization of the learned autoregressive distribution towards the empirical distribution of the training data. Can the authors comment on whether this is a reasonable or fair characterization? Can they comment on how the NCE training is expected to result in an energy function that prefers globally coherent / plausible sequences?

3. Can the authors address the third bullet point under "Weaknesses" - specifically, whether the baseline autoregressive models also use "best-of-$M$" decoding, and if not, why not? My main concern is that taking the best of $M$ autoregressive generations is an advantage on its own, for a potentially wide variety of choices as to how we define "best," and it would be informative for readers to be able to compare the HYPRO method here (which uses the energy function to define "best") against some baseline.

Minor / misc.:

1. It seems like Alg. 1 is just the standard thinning algorithm for simulation of a non-homogeneous (multivariate) point process. Could the authors clarify if there's a novel contribution here?


**Limitations:**

The authors have adequately addressed limitations and potential impacts.

**Strengths And Weaknesses:**

Strengths.

- This is a well-written paper. The motivation, background, method, and experimental analysis are informative and presented in a lucid style. I would rate this paper highly in terms of "clarity."

- The experiments are plausible and involve a thorough analysis of the various modeling degrees of freedom in the proposed approach, even those with modest or inconclusive results. The authors study the performance of the HYPRO method over multiple time horizons, quantify uncertainty in the results, and are able to show consistent improvement over their baselines. They provide code for all experiments.

Weaknesses.

- The authors repeatedly claim that autoregressive models fail at long-horizon prediction due to the problem of compounding / cascading errors. This claim seems plausible, but there is essentially no evidence provided to support it;  $\S$2.2., for example, has no references. Given that the premise of this paper is to "fix" that problem, it would be useful to see a stronger argument or evidence here.

- The energy function is trained by minimization of a noise-contrastive loss, where the "noise" is distribution is just that of the stage-one autoregressive model. This seems to encode the assumption that *every* generation of the autoregressive model is bad, a much stronger (and less plausible) claim than "autoregressive generations are prone to cascading errors." It is not clear to me how the "contrast" established in this noise-contrastive loss could lead the energy function to learn some notion of global coherence or plausibility of sequences.

- Generation from the HYPRO model involves selecting the "best" (according to the energy function-based weights) among $M$ generated sequences from the autoregressive model. This raises some questions for fair comparison to the autoregressive baselines. First, it's not clear whether this "best-of-$M$" decoding method is also extended to the baseline autoregressive models, or (assuming it is not) how much better these baselines would perform if it had been. Second, there don't appear to be any ablations that attempt to isolate the impact of using *specifically* the learned energy function to select among the $M$ best baselines vs. some baseline criterion.

---

> ### Author Response · Authors · 2022-08-02
> **point-to-point reply 3**
>
> > Generation from the HYPRO model involves selecting the "best" (according to the energy function-based weights) among  generated sequences from the autoregressive model. This raises some questions for fair comparison to the autoregressive baselines. First, it's not clear whether this "best-of-" decoding method is also extended to the baseline autoregressive models, or (assuming it is not) how much better these baselines would perform if it had been.
> > Second, there don't appear to be any ablations that attempt to isolate the impact of using specifically the learned energy function to select among the  best baselines vs. some baseline criterion.
> > Can the authors address the third bullet point under "Weaknesses" - specifically, whether the baseline autoregressive models also use "best-of-" decoding, and if not, why not? My main concern is that taking the best of autoregressive generations is an advantage on its own, for a potentially wide variety of choices as to how we define "best," and it would be informative for readers to be able to compare the HYPRO method here (which uses the energy function to define "best") against some baseline.
>
> We followed your advice to extend the ''best-of-'' decoding method to the autoregressive base model. The results are in [New Results - New Baseline]. It turns out that this new baseline method underperforms our methods and other baselines.
>
> Note that the DualTPP baseline method uses macro statistics as their ''best-of-'' criterion and it also underperforms our methods.
>
> Should you have suggestions of other criteria to try, we'd be more than happy to try them and report the results.
>
> > It seems like Alg. 1 is just the standard thinning algorithm for simulation of a non-homogeneous (multivariate) point process. Could the authors clarify if there's a novel contribution here?
>
> It calls the thinning algorithm (Alg-1:ln-4 to ln-14) N times to draw N continuations. In the final version, we will write a separate algorithm for the thinning algorithm (perhaps moved to Appendices) and call that algorithm in Alg-1. That will save space for our new results.

---

> ### Author Response · Authors · 2022-08-02
> **point-to-point reply 2**
>
> > The details of the NCE training of the energy function lead me to wonder whether the second stage of the HYPRO model is really just enforcing an after-the-fact regularization of the learned autoregressive distribution towards the empirical distribution of the training data. Can the authors comment on whether this is a reasonable or fair characterization? Can they comment on how the NCE training is expected to result in an energy function that prefers globally coherent / plausible sequences?
>
> Long in short: the energy function fits the residuals that are not explained by the autoregressive model itself. This remark holds regardless of the training methods.
>
> Moreover, globally-normalized models naturally consider global features of sequence data regardless of the training methods---that is an advantage of energy-based models over autoregressive models. This comparison has been extensively studied in literature like:
> Characterizing and Overcoming the Limitations of Neural Autoregressive Models. Goyal, Dissertation 2021.
>
> As for ''how NCE training helps an energy function achieve this capacity'' in particular: NCE learns to discriminate the true sequence---as a whole---against the noise sequences that are drawn from the autoregressive model.
> Specifically, the energy function treats the ground-truth sequences as the ''better'' ones (i.e., positive samples) and assign low energy values to them; it learns to assign higher energies to the noise. Since the noise is drawn from the trained autoregressive model and already has decent quality, discriminating true against noise will push the energy function to pick up the higher-level features of sequences (e.g., A co-occur with C even if there are many other events in-between) that may have been missed by the autoregressive base model.

---

> ### Author Response · Authors · 2022-08-02
> **point-to-point reply 1**
>
> Thank you for your constructive feedback. We appreciate it and have added new experiment results to resolve your concerns; see [New Results]. Hope you like them.
>
> > The authors repeatedly claim that autoregressive models fail at long-horizon prediction due to the problem of compounding / cascading errors. This claim seems plausible, but there is essentially no evidence provided to support it; §2.2., for example, has no references. Given that the premise of this paper is to "fix" that problem, it would be useful to see a stronger argument or evidence here.
> > Can the authors provide some citations or evidence for compounding errors in long-horizon event forecasting?
>
> We have provided empirical evidence for cascading errors in [New Results - Cascading Errors]. Does it look reasonable to you?
>
> Also, please do not miss that: in Analysis-IV (line-263), we have already shown that our method has a larger advantage (and baseline has larger drop) when the horizon is longer.
>
> There is indeed previous literature that studies this problem and we cited them in the Related Work section in our paper. We will move some of them upfront to the Introduction section to clarify any confusion. E.g.,
> Long Horizon Forecasting With Temporal Point Processes, Deshpande et al, WSDM, 2021.
> Residual Energy-Based Models for Text Generation, Deng et al, ICLR, 2020.
> Sequence level training with recurrent neural networks. Ranzado et al, ICLR 2016.
> Characterizing and Overcoming the Limitations of Neural Autoregressive Models. Goyal, Dissertation 2021.
>
> > The energy function is trained by minimization of a noise-contrastive loss, where the "noise" is distribution is just that of the stage-one autoregressive model. This seems to encode the assumption that every generation of the autoregressive model is bad, a much stronger (and less plausible) claim than "autoregressive generations are prone to cascading errors." It is not clear to me how the "contrast" established in this noise-contrastive loss could lead the energy function to learn some notion of global coherence or plausibility of sequences.
>
> First, using NCE does not mean every sample from the noise distribution (i.e., the autoregressive base model) is bad. Actually, NCE relies on the noise distribution being decent to work well in practice---that is why the noise distribution is often trained; this has been discussed in a series of papers on NCE such as:
> Noise-contrastive estimation: A new estimation principle for unnormalized statistical models. Gutmann and Hyvarinen 2010.
> Noise-Contrastive Estimation for Multivariate Point Processes. Mei et al. NeurIPS 2020.
>
> We just adopted the common practice: we used the trained autoregressive model as our noise distribution since it has already been decent in fitting the data.
>
> Second, it is not that ''noise-contrastive'' leads to ''energy function''.
> Instead, it is the "energy function" that leads to ''noise-contrastive'' because it can not be trained by MLE (line-83 to line-86).
>
> It has been well-known that energy functions are better at learning global structures, which is one of its key modeling advantages over autoregressive models. Below is a (far-from-exhaustive) list of papers that have discussed this advantage:
> Trajectory Prediction with Latent Belief Energy-Based Model, Pang et al, CVPR 2021.
> Residual Energy-Based Models for Text Generation, Deng et al, ICLR 2020.
> Implicit Generation and Modeling with Energy-Based Models, Du et al, NeurIPS, 2019.
> Training Energy-Based Models for Time-Series Imputation, Brakel et al, JMLR, 2013.
> We will cite these papers in our final version.

---

> ### Comment · Reviewer_XUS4 · 2022-08-05
> **reply to author discussion**
>
> Thanks to the authors for their extensive discussion.
>
> My main concerns involved (1) evidence for the “cascading errors” problem, (2) justification for the NCE training scheme with the phase-1 autoregressive model as the “noise” distribution, and (3) distinguishing between potential gains from generic “best-of-M” generation versus “best-of-M” with “best” specifically determined by the energy function.
>
> The authors have addressed (1) by a combination of references and some further investigation on their own data. They have addressed (2) through some clarifying discussion. They have addressed (3) by further experiments involving “best-of-M” generation with “best” defined as probability under the phase-1 autoregressive model. I’m somewhat surprised to see that these results are so weak for the autoregressive model - they seem to suggest that the model is quite poorly calibrated even after training. Nonetheless I think these results provide helpful evidence that the energy function approach adds value beyond a generic “best-of-M” generation scheme, and in implicitly highlighting the poor calibration of the autoregressive model they also relieve some of my concerns regarding (2).
>
> In view of these developments, I’m prepared to raise my score for this submission. If this submission is accepted, I would encourage the authors to include some of the further experimental results and discussion here into the final version.

---

> > ### Author Response · Authors · 2022-08-06
> > **thank you and we will surely include all the new experiments and discussion**
> >
> > Thank you!
> >
> > We will surely include all the rebuttal content (new results and discussion) in the final version.
> >
> > The rebuttal seems long, but an extra page will be allowed for the camera-ready version per NeurIPS guidelines.
> > (See https://nips.cc/Conferences/2022/CallForPapers)
> > Additionally, we can:
> > 1. trim some existing content: e.g., many lines in Alg-1 can be a separate algorithm box and moved to Appendices
> > 2. for many (old and new) results, describe them concisely in the main paper but leave details to Appendices
> >
> > So it is definitely feasible to include all the new content in our final version.

---

### Official Review · Reviewer_WY8n · 2022-07-11

**Rating:** 6
**Confidence:** 3
**Soundness:** 2 fair
**Presentation:** 4 excellent
**Contribution:** 2 fair

**Summary:**

This paper presents HYPRO, a novel model for long-horizon prediction of event sequences. HYPRO its a hybrid model, first an autoregressive model learns to propose predictions, and a second energy function weights them to produce more realistic final predictions. Two auto autoregressive models are used, NHP and AttNHP, and a continuous-time Transformer is used as the energy function. HYPRO can be trained with two losses: a binary-NCE objective and a multiple-NCE objective. Authors also propose a novel regularization to enforce larger distances between true and noise sequences. The paper compares HYPRO against baselines on two benchmark datasets.

**Questions:**

- Why were other benchmark datasets not included in the experiments? The AttNHP paper [1] uses MIMIC-II, StackOverflow, RoboCup, and IPTV datasets, some of which have longer time between events which would be more adequate to test the proposed model on the long-horizon setting.
- How does training and inference time compare against baselines?

**Limitations:**

Limitations are societal impacts are addressed in the paper.

**Strengths And Weaknesses:**

Strengths:
- The hybrid approach proposed in the paper was not used before for the prediction of event sequences.
- Authors propose a novel regularization technique to enforce larger distances between sequences.
- Authors test the model with multiple training objectives, results confirm the superior properties of the multi-NCE objective.
- HYPRO achieves better performance than baselines on two benchmark datasets.
- The experimental details are explained in detail in the appendix.
- The paper is well written and clear.

Weaknesses:
- The core idea of HYPRO was already studied in several previous works for text generation. The paper's novelty is limited to applying it to a different task and using different architectures.
- The improvements of the novel regularization technique are not statistically significant in any of the datasets.
- While the paper's primary focus is long-horizon predictions, the experiments on the prediction horizon are weak. The authors only include the analysis on one dataset. The longest horizon on the main results seems short (6 hours) compared to the motivation problems stated in the introduction (several months).
- One of the motivations for the training objectives and inference method is the "low computational cost"; however, the experiments do not evaluate training times.
- Limited benchmark datasets.

---

> ### Author Response · Authors · 2022-08-02
> **Thank you for your feedback.**
>
> Thank you for your constructive feedback. We have added new experiment results to address your concerns. In this message, we'd like to respond to your remaining points.
>
> > While the paper's primary focus is long-horizon predictions, the experiments on the prediction horizon are weak. The authors only include the analysis on one dataset. The longest horizon on the main results seems short (6 hours) compared to the motivation problems stated in the introduction (several months).
>
> The concept of ''long-horizon'' is relative to ''only predicting the next event token''. Please see line-22 to line-25: the core is to ''predict the series of future events over a given time interval''. In our experiments, we predict 20 future events which is much more than ''just the next event''.
>
> Moreover, what matters more in practice is the number of events to predict, not the length of the time interval, since the latter can be made easy by uniformly scaling all the time intervals: if ''6 hours'' in data A and ''6 months'' in data B both have 20 events on average, then they are the same difficult to a model since one could scale the time intervals such that ''6 hours'' in data A and ''6 months'' in data B are both ''time unit 1''.
>
> > The core idea of HYPRO was already studied in several previous works for text generation. The paper's novelty is limited to applying it to a different task and using different architectures.
>
> What we apply to is not ''yet another task in text/NLP'' but another domain where data is significantly different: i.e., continuous-time vs. discrete-time.
> Do our arguments in [Technical Contribution] sound reasonable to you?
>
> > The improvements of the novel regularization technique are not statistically significant in any of the datasets.
>
> The improvements are significant under the paired permutation test in half of the cases. Does our discussion in [Statistical Significance] address your concern?
>
> > One of the motivations for the training objectives and inference method is the "low computational cost"; however, the experiments do not evaluate training times.
> > How does training and inference time compare against baselines?
>
> Please see [New Results - Computation Cost] for details about training and inference cost.
>
> More importantly, we propose the specific training and inference methods not because they are faster but because they make the problem tractable. Note that typical MLE training for our proposed model is intractable since the partition function Z (eq-2) sums infinitely many terms: we discussed this intractability in the paper; see line-83 to line-86.
>
> > Limited benchmark datasets.
> > Why were other benchmark datasets not included in the experiments? The AttNHP paper [1] uses MIMIC-II, StackOverflow, RoboCup, and IPTV datasets, some of which have longer time between events which would be more adequate to test the proposed model on the long-horizon setting.
>
> We added results on new data: i.e., StackOverflow. We also explained why we didn't use the other datasets. Please see [New Results - New Dataset].
>
> Please let us know if our response has addressed your concerns. We'd be happy to have more discussion and improve our paper accordingly. Thank you.

---

> > ### Comment · Reviewer_WY8n · 2022-08-08
> > **Response**
> >
> > Thank you for your detailed response and new results, they clarify some of my initial concerns, I will raise my initial score.

---

> > > ### Author Response · Authors · 2022-08-08
> > > **Thank you.**
> > >
> > > We thank you for your supportive feedback.
> > >
> > > We will include all the new discussion and new results in our final version.

---

### Author Response · Authors · 2022-08-02
**New Results - Ablation**

[Ablation: Number of Negative Samples in Training]

On the Taobao dataset, we did an ablation study for the number of negative samples during training. We used hypro-a-b without our proposed regularization. Here are the results on the test set.
| # of negative samples | RMSE | OTD |
| ----------- | ----------- |----------- |
| 1 | 9.935 (+/- 0.019) | 36.410 (+/- 0.284) |
| 5 | 9.918 (+/- 0.020) | 36.223 (+/- 0.291) |
| 10 | 9.922 (+/- 0.023) | 36.229 (+/- 0.276) |

As we can see, increasing the number of negative samples from 1 to 5 has brought significant improvements but further increasing it to 10 does not. Throughout the paper, we used 5 (see Appendix B.3).

[Ablation: Number of Samples in Eval]

We did an ablation study for the number of samples used in inference. Here are the results.
The model is hypro-a-b, without our proposed regularization.

Taxi Dataset
| # of samples | RMSE | OTD |
| ----------- | ----------- |----------- |
| 5 | 4.185 (+/- 0.015) | 20.132 (+/- 0.182) |
| 20 | 4.112 (+/- 0.014) | 19.729 (+/- 0.180) |
| 40 | 4.110 (+/- 0.017) | 19.725 (+/- 0.197) |

Taobao Dataset
| # of samples | RMSE | OTD |
| ----------- | ----------- |----------- |
| 5 | 9.965 (+/- 0.019) | 37.710 (+/- 0.284) |
| 20 | 9.918 (+/- 0.020) | 36.223 (+/- 0.291) |
| 40 | 9.917 (+/- 0.028) | 36.275 (+/- 0.290) |

As we can see, the results are improved when we increase the number of samples from 5 to 20, but they stop improving when we further increase it.

---

### Author Response · Authors · 2022-08-02
**New Results - New Dataset**

[New Dataset]

Reviewer WY8n suggested that we evaluate on the datasets that Yang et al. 2022 used (i.e., MIMIC-II, StackOverflow, RoboCup, and IPTV).

We conducted new experiments on the StackOverflow data.
The number of event tokens in training, dev, and test are 91K, 26K, and 27K, respectively. The average length is 65. The dataset is available at https://drive.google.com/file/d/1YOaOukggcEL5boWatmWggwzJUPy7gle_/view?usp=sharing

| method \ metric | RMSE | OTD |
| ----------- | ----------- |----------- |
| DualTPP | 5.508 (+/- 0.021) | 39.265 (+/- 0.061) |
| NHP | 6.011 (+/- 0.016) | 39.725 (+/- 0.064) |
| NHP-lg | 5.598 (+/- 0.016) | 39.593 (+/- 0.049) |
| AttNHP | 5.340 (+/- 0.018) | 39.611 (+/- 0.065) |
| AttNHP-lg | 5.351 (+/- 0.019) | 39.391 (+/- 0.052) |
| hypro-a-b | 5.303 (+/- 0.021) | 38.371 (+/- 0.035) |
| hypro-a-m | 5.266 (+/- 0.024) | 38.188 (+/- 0.038) |

Our methods (hypro-a-b and hypro-a-m) still significantly outperform all the other methods on this dataset.

The other suggested datasets are not a good fit for our paper.
The sequences of MIMIC-II are too short because it is ICU data.
RoboCup and IPTV have too few sequences for training the energy function. Note that these datasets were a good fit for Yang et al. 2022 since they require temporal logic to model well and that is what Yang et al. 2022 is concerned about.

Reference:
Transformer Embeddings of Irregularly Spaced Events and Their Participants
Chenghao Yang, Hongyuan Mei, Jason Eisner
ICLR 2022

---

### Author Response · Authors · 2022-08-02
**New Results - Computation Cost**

[Computation Cost]

Our experiments are run on a machine with one single NVIDIA Tesla P100 GPU. Our device details are in Appendix B.3. (Appendix B.3 made a mistake of saying we used ``2 GPUs''. We'll correct it in the final version.)

For training, our batch size is 32. For Taobao and Taxi dataset, training the baseline NHP, NHP-lg, AttNHP, AttNHP-lg approximately takes 1 hour, 1.3 hour, 2 hours, and 3 hours, respectively (12, 16, 25, 38 milliseconds per sequence), training the continuous-time LSTM energy function and continuous-time Transformer energy function takes 20 minutes and 35 minutes (4 and 7 milliseconds per sequence pair) respectively.

For inference, energy functions take ~2--4 milliseconds for inference. It takes ~0.2 seconds to draw a sequence from the autoregressive base model. The submitted implementation takes 20 x 0.2 seconds for hypro-a-b and hypro-a-m to draw 20 sequences for each sequence prefix, since that implementation doesn't parallelize the thinning algorithm. However, during rebuttal, we have managed to upgrade our thinning algorithm implementation so it can now draw multiple sequences at a time in parallel: now it takes only ~0.4 seconds to draw 20 sequences---twice as drawing a single sequence. We will release this new implementation once the paper is published.

---

### Author Response · Authors · 2022-08-02
**New Results - New Baseline**

[New Baseline]

Reviewer XUS4 and Reviewer urPE are concerned whether we have made a fair comparison to the autoregressive base model. To address this concern, we developed a new baseline method (suggested by urPE): we draw continuations from the autoregressive base model, rank the completed sequences based on their probabilities under the base model, and then select the one with highest probability; this is similar to our proposed HYPRO framework except that the scorer is the base model itself.

We trained and evaluated this new baseline on the Taobao dataset. Here are the results of using AttNHP as the base model.
| method \ metric | RMSE | OTD |
| ----------- | ----------- |----------- |
| New baseline | 10.87 (+/- 0.029) | 36.254 (+/- 0.301) |
| AttNHP | 10.050 (+/- 0.025) | 37.456 (+/- 0.243) |
| AttNHP-lg | 10.015 (+/- 0.021) | 37.398 (+/- 0.262) |
| hypro-a-b | 9.918 (+/- 0.020) | 36.223 (+/- 0.291) |
| hypro-a-m | 9.690 (+/- 0.023) | 36.486 (+/- 0.222) |

As we can see, this new method is actually worse than the other baseline methods.

Please note that the base model is not the only baseline method that we compared to. We also compared to DualTPP, a model that was developed to address the same ``long-horizon prediction'' problem and ranks sequences based on their macro statistics. Our methods perform much better; see Figure-1 and the related discussion.

---

### Author Response · Authors · 2022-08-02
**New Results - Statistical Significance**

[Statistical Significance]

Reviewer WY8n is concerned with the significance of our proposed regularization technique. That may be because some error bars in our figures overlap. Please note: while non-overlapped error bars imply statistical significance, overlapped error bars do not necessarily mean non-significance---we just need to perform a more rigorous significance test in those cases.

We used the paired permutation test. See details at: https://axon.cs.byu.edu/Dan/478/assignments/permutation_test.php

We tested the results in Figure-3 and it turns out that the performance differences are strongly significant for hypro-a-b (p-value $<0.05$) and weakly significant for hypro-a-m (p-value $\approx 0.1$). This is consistent with the findings in Analysis-III and Figure-2 in the paper.

p-values of perm test on Taobao Data
| method \ metric | RMSE | OTD |
| ----------- | ----------- |----------- |
| hypro-a-b | 0.0336 | 0.0363 |
| hypro-a-m | 0.0987 | 0.100 |

p-values of perm test on Taxi Data
| method \ metric | RMSE | OTD |
| ----------- | ----------- |----------- |
| hypro-a-b | 0.0452 | 0.0487 |
| hypro-a-m | 0.102 | 0.096 |

---

### Author Response · Authors · 2022-08-02
**New Results - Cascading Errors**

[Cascading Errors]

Review XUS4 is concerned with whether the problem of ``cascading errors'' actually exists. We investigated the existence of the issue empirically on the Taobao dataset.

We first investigate whether the event type prediction errors are cascaded through the subsequent events. To do this, we grouped the sequences based on how early the first event type prediction error was made and then compared the event type prediction accuracies on the subsequent events. The results are in the table below:
| first error made in ?-th event token | base model AttNHP | our hybrid model hypro-a-b |
| ----------- | ----------- |----------- |
| 1st      | 28.716% | 33.012% |
| 5th   | 41.125% | 48.107% |
| 10th   | 55.516% | 64.417% |

As we can see, when mistakes are made earlier in the sequences, we tend to end up with lower accuracies on the subsequent predictions; that means event type prediction errors are indeed cascaded through the subsequent predictions.
Moreover, in each group, our hybrid model enjoys a higher prediction accuracy; that means it indeed helps mitigate this issue.

We then investigate whether the event time prediction errors are cascaded. For this, we performed a linear regression: the independent variable $x$ is the absolute error of the prediction on the time of the first event token; the dependent variable $y$ is the averaged absolute error of the prediction on the time of the subsequent event tokens.
Our fitted linear model is $y = 0.7965 x + 0.3219$ where the p-value of the coefficient of $x$ is $\approx 0.0001 < 0.01$. That means: time prediction errors are indeed cascaded through the subsequent predictions.

---

### Author Response · Authors · 2022-08-02
**Technical Contribution**

[Technical Contribution]

Reviewer WY8n is concerned with the novelty this paper since ``the core idea was already studied in several previous works for text generation.''

Isn't it a technical contribution to (be the first to) extend such a general technical idea to the time-stamped event sequences? Such data is ubiquitous in real-world applied domains (e.g., finance, medicine); long-horizon prediction of such data is a critical task but understudied. No hybridly-normalized neural probabilistic model exists for such data. So why not have one? We presented a clean and viable framework for others to use, with the best results to date and an easy-to-use codebase. This new paper will significantly facilitate future research in this area.

Moreover, our paper indeed has new technical ingredients: we leveraged different model architectures (continuous-time LSTM/Transformer) and proposed new training objectives (eq-4,5). The architectural changes are necessary to handle continuous time and the training modifications improve the results in practice.

Reviewer urPE suggests that we'd be upfront about our technical contributions: i.e., ``the first to adapt the hybrid model to event sequence data with XYZ architectural + training modifications''. We will surely take this advice and rephrase our introduction (and anywhere else appropriate).

Furthermore, we have been careful in evaluating the model and carried out a thorough empirical investigation---this is also our technical contribution. In this response, we have included a significant amount of new results to answer your questions about the experiments; see New Results. We will include them in our final version.

---

### Author Response · Authors · 2022-08-02
**Thank you for the constructive feedback**

We thank all the reviewers for their constructive feedback.
In this response, we will first use a few reply-to-all messages to clarify our technical contribution and present a variety of new experiment results. Then we will provide a point-to-point response to each individual review.
We can surely consolidate all the paper improvements (including clarified technical contributions, new results, new discussion, etc) in our final version.

Hope that our response will have your minds reassured.

---

### Author Response · Authors · 2022-08-08
**Thank you and we will structure the final version accordingly.**

We thank all the reviewers for their supportive feedback.

We will surely structure the final version as we have discussed and agreed. Most importantly, we will rewrite the Introduction to emphasize our precise technical contributions (see Technical Contribution in our rebuttal).

We will also include all the new results in the final version. There is a lot of new content, but the NeurIPS camera-ready allows an extra page. Additionally, we will:
1. trim some existing content: e.g., many lines in Alg-1 can be a separate algorithm box and moved to Appendices
2. for many (old and new) results, describe them concisely in the main paper but leave details to Appendices

---

### Meta-Review · Area_Chair_EtQX · 2022-08-26

**Recommendation:** Accept
**Confidence:** Certain

**Metareview:**

The authors propose in this submission a hybrid approach for the event prediction problem combining an auto-regressive model and an energy model aiming to correct a compounding error issue that auto-regressive models can have. Reading the submission and according to reviewers,  paper is well written and the approach is reasonable and well-motivated. Moreover, the discussion with the reviewers has allowed the authors to provide additional arguments supporting their different claims convincingly.
Therefore I recommend this paper for acceptance.

**Award:**

No

---

### Decision · Program_Chairs · 2022-09-14

Accept